# Incidence of macular hole in patients undergoing pars plana vitrectomy for submacular hemorrhage

**Riko Matsumoto** [ORCID]*, **Masashi Kakinoki, Osamu Sawada, Shumpei Obata, Yoshitsugu Saishin, Masahito Ohji**

Department of Ophthalmology, Shiga University of Medical Science, Otsu, Shiga, Japan

* rokamoto@belle.shiga-med.ac.jp

## Abstract

### Purpose

To investigate the incidence of macular hole (MH) during pars plana vitrectomy (PPV) for submacular hemorrhage (SMH) due to either retinal arterial macroaneurysm (RAM) rupture or age-related macular degeneration (AMD).

### Methods

We retrospectively evaluated 47 eyes of 47 patients with SMH due to RAM rupture or AMD who underwent PPV. The presence or development of MHs was confirmed intraoperatively by the surgeon. We compared the incidence of MH between the RAM and AMD groups.

### Results

In the RAM group, a MH was found in five of 28 (17.9%) eyes, and a MH was developed in three eyes by the surgical procedure. In the AMD group, there was no MH. All MHs were identified intraoperatively and closed by the same surgical procedure. In the RAM group, subretinal tissue plasminogen activator (t-PA) injection was performed in 12 eyes. Of the 12 eyes, two developed an MH before t-PA subretinal injection. Of the remaining 10 eyes, four (40%) developed an MH intraoperatively or postoperatively.

### Conclusion

MHs are complications of PPV for SMH that occur more frequently in patients with SMH due to RAM rupture than in patients with SMH due to AMD.

## Introduction

Submacular hemorrhage (SMH) is a condition that causes sudden vision loss and often results from retinal arterial macroaneurysm (RAM) rupture or age-related macular degeneration (AMD). Macular hemorrhages require rapid removal or displacement because they can produce irreversible iron- and fibrin-induced damage in outer layers of the retina [1,2].

**Data availability statement:** All relevant data are within the manuscript and its Supporting Information files.

**Funding:** The author(s) received no specific funding for this work.

**Competing interests:** The authors have declared that no competing interests exist.

Treatments for hemorrhage removal comprise pneumatic displacement with gas or pars plana vitrectomy (PPV). SMH can be treated by pneumatic displacement with gas or gas plus tissue plasminogen activator (t-PA) [3,4], but PPV with subretinal t-PA injection is preferred because it enables more reliable displacement. Furthermore, PPV is necessary in cases of SMH with sub-internal limiting membrane (ILM) hemorrhage or vitreous hemorrhage (VH).

Macular holes (MHs) reportedly occur in combination with SMH due to RAM rupture or AMD [5–8]. The presence of an MH is a factor associated with poor visual acuity after SMH displacement [9]. MHs can occur at various times: before surgery, during surgery, and after surgery [10]. To our knowledge, no studies have directly compared the incidence of MH due to SMH among underlying diseases. Therefore, this study retrospectively compared the incidence of MH in patients who underwent PPV for SMH between patients with SMH due to RAM rupture and patients with SMH due to AMD.

## Patients and methods

This retrospective study included 47 patients with SMH due to RAM rupture or AMD who underwent PPV between April 2010 and June 2023 at the Shiga University of Medical Science Hospital. We excluded patients who had undergone pneumatic displacement of SMH before PPV. The study adhered to the tenets of the Declaration of Helsinki. The Shiga University of Medical Science Research Review Board approved the study protocol. Informed consent was obtained in the form of opt-out on the web-site. We accessed the data for research purposes in 25/08/2022-05/09/2022 and 27/08/2023. We have created anonymized information by giving research subjects research IDs that do not identify individuals on their own. The information obtained in the research is anonymized by managing it using the research ID.

Our indications for surgery in SMH are cases complicated by VH, preretinal hemorrhage or sub-ILM hemorrhage and cases where the size of the SMH is approximately one disc diameter or larger.

We retrospectively retrieved the following data from patients' records: age, sex, best-corrected visual acuity (BCVA) (preoperatively and 1 month postoperatively), days from awareness of onset to surgery, preoperative VH, intraoperative t-PA use, ILM peeling involving the fovea, and Mention of MH in the surgical records. VH was determined based on fundus photographs and medical records. The presence or development of MH was confirmed by preoperative or postoperative optical coherence tomography (OCT) or intraoperatively by the surgeon. Surgeries were performed by five surgeons (M.O., M.K., O.S., H.K., and R.M.). The use of t-PA was determined at the surgeon's discretion; t-PA was administered by preoperative intravitreal injection or by intraoperative subretinal injection. The surgical technique for subretinal injection of t-PA is as follows: the ILM at the planned t-PA injection site was peeled, and BSS was injected into the subretinal space using a micro cannula, after which 0.1 ml (25μg) of t-PA was injected. BSS was injected before tPA injection to inject the exact amount of tPA. The injection pressure was controlled using a viscous fluid control system (Alcon Laboratories, FortWorth, TX). ILM peeling was also performed at the surgeon's discretion. All patients underwent fluid-air exchange at the end of surgery, followed by $SF_6$ or air tamponade. Patients were instructed to remain in the prone position for 1–2 weeks postoperatively. We compared the incidence of MH in patients who underwent PPV for SMH between the RAM and AMD groups.

BCVA values measured by the Landolt C chart test were converted to logarithm of the minimum angle of resolution (logMAR) equivalents for statistical analysis. Based on a previous report, counting fingers was converted to a logMAR value of 1.85 and hand motion was converted to a logMAR value of 2.30 [11]. All statistical analyses were performed with EZR

(Saitama Medical Center, Jichi Medical University, Saitama, Japan), a graphical user interface for R (The R Foundation for Statistical Computing, Vienna, Austria) [12]. The Mann–Whitney U test was used to compare the RAM and AMD groups. Fisher's exact test was used to compare the number of eyes with MH between the RAM and AMD groups. Data (age, logMAR BCVA, and days from onset) are expressed as means (standard deviations: SD). P values <0.05 were considered statistically significant.

## Results

This study included 28 eyes of 28 patients with SMH due to RAM rupture and 19 eyes of 19 patients with SMH due to AMD. The type of AMD was typical AMD in six eyes, polypoidal choroidal vasculopathy in twelve eyes, and retinal angiomatous proliferation in one eye. One patient in the RAM group and one patient in the AMD group were excluded from the study because they had undergone pneumatic displacement of SMH before PPV. The patients' characteristics and surgical outcomes are presented in Table 1. The RAM group comprised 22 women and six men, whereas the AMD group comprised two women and 17 men (p < 0.01). The mean ages and standard deviations were 82.9 (SD 6.0) years (range, 67–91) in the RAM group and 73.2 (SD 8.5) years (range, 53–87) in the AMD group (p < 0.01). The mean times from onset to surgery were 9.6 (SD 7.4) days in the RAM group and 10.1 (SD 7.0) days in the AMD group; these did not significantly differ (p = 0.60). There were no significant differences in logMAR BCVA preoperatively or 1 month postoperatively between the two groups (preoperative: p = 0.06, postoperative: p = 0.55). Both groups showed significant improvement in BCVA after surgery (p < 0.01 for both analysis). Preoperative VH was observed in seven eyes (25.0%) in the RAM group and 10 eyes (52.6%) in the AMD group (p = 0.07). In the RAM group, intravitreal t-PA injection was performed in three eyes (10.7%); the day before surgery in two eyes, and during surgery in the other eye. In the AMD group, intravitreal t-PA injection was performed in two eyes (10.5%) the day before surgery. Subretinal t-PA injection was performed in 12 eyes (42.9%) in the RAM group and nine eyes (47.4%) in the AMD group (p = 0.77). ILM peeling at the central fovea was performed in 21 eyes (75.0%) in the RAM group and two eyes (10.5%) in the AMD group (p < 0.01). All data are provided in the S1 Dataset. The mean and median postoperative observation periods for the RAM group were found to be 13.3 and 12 months, respectively. The mean and median postoperative observation periods for the AMD group were found to be 40.9 and 37 months, respectively.

**Table 1. Baseline characteristics and surgical outcomes.**

|  | RAM (n = 28) | AMD (n = 19) | P |
|---|---|---|---|
| Female/male | 22/ 6 | 2/ 17 | < 0.01 |
| Age, years (SD) | 82.9 (6.0) | 73.2 (8.5) | < 0.01 |
| Preoperative BCVA, logMAR (SD) | 1.49 (0.40) | 1.70 (0.54) | 0.06 |
| 1M postoperative BCVA, logMAR (SD) | 1.07 (0.48) | 1.19 (0.55) | 0.55 |
| Days from onset (SD) | 9.6 (7.2) | 10.1 (7.0) | 0.03 |
| Preoperative vitreous hemorrhage (%) | 7 (25.0) | 10 (52.6) | 0.07 |
| Intravitreal t-PA injection, eyes (%) | 3 (10.7) | 2 (10.5) | 1 |
| Subretinal t-PA injection, eyes (%) | 12 (42.9) | 9 (47.4) | 0.77 |
| ILM peeling involving fovea, eyes (%) | 21 (75.0) | 2 (10.5) | < 0.01 |

RAM: retinal artery macroaneurysm, AMD: age-related macular degeneration

BCVA: best-corrected visual acuity, ILM: internal limiting membrane.

In the RAM group, an MH was found in five of 28 eyes, and an MH was developed in three cases by the surgical procedure. The incidence of MH due to the RAM itself was 17.9%, and the incidence of MH related to the surgical procedure (subretinal t-PA injection) was 13.0%. In the AMD group, there was no MH (p < 0.01).

All MHs were identified intraoperatively and none were found by OCT. Table 2 shows baseline characteristics and surgical outcomes between the RAM patients who acquired and did not develop MH. ILM peeling involving the fovea was performed in six cases of the eight cases that developed MH. Subretinal t-PA injection was performed in four cases of these six cases. The times at which MHs were found or developed are presented in Fig 1. One MH was found immediately after core vitrectomy, but it was unclear whether the MH had occurred preoperatively or intraoperatively because the patient exhibited VH before the surgery. An MH was found during or immediately after ILM peeling in three eyes. An MH

**Table 2. baseline characteristics and surgical outcomes between RAM patients who acquired and did not develop MH.**

|  | MH(+) (n = 8) | MH(-) (n = 20) | P |
|---|---|---|---|
| **female/male** | 6/ 2 | 16/ 4 | 0.28 |
| **Age, years (SD)** | 80.8 (7.6) | 83.8 (4.8) | 0.51 |
| **Preoperative BCVA, logMAR (SD)** | 1.56 (0.39) | 1.46 (0.40) | 0.61 |
| **1M postoperative BCVA, logMAR (SD)** | 0.98 (0.51) | 1.11 (0.47) | 0.37 |
| **Days from onset (SD)** | 8.4 (2.8) | 10.2 (8.3) | 0.59 |
| **Preoperative vitreous hemorrhage (%)** | 2 (25.0) | 5 (25.0) | 1 |
| **Intravitreal t-PA injection, eyes (%)** | 0 | 3 (15.0) | 0.54 |
| **Subretinal t-PA injection, eyes (%)** | 6 (75.0) | 6 (30.0) | 0.04 |
| **ILM peeling involving fovea, eyes (%)** | 6 (75.0) | 15 (75.0) | 1 |

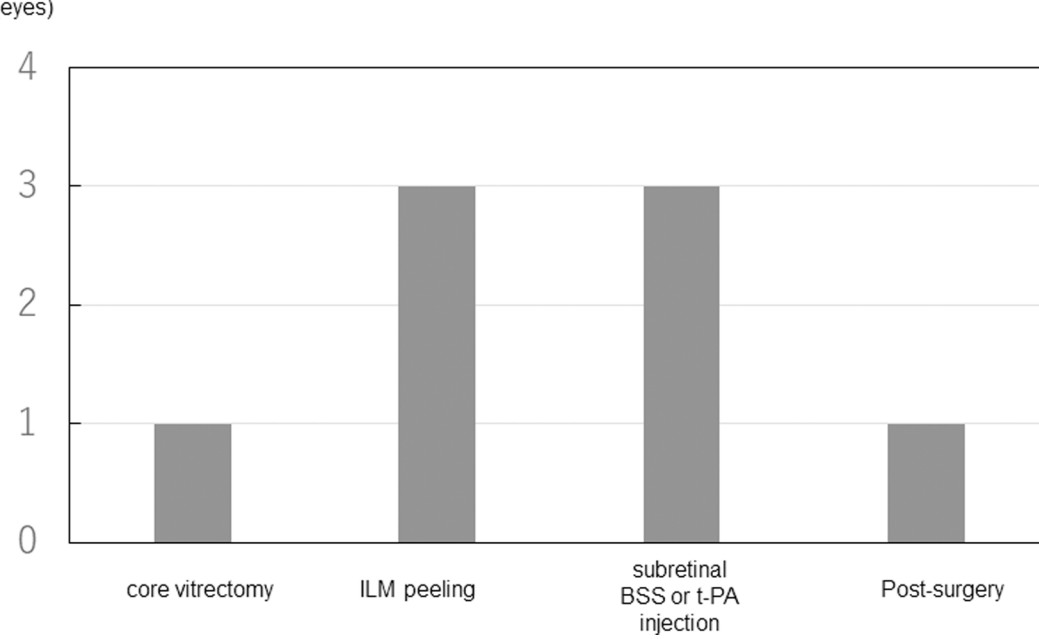

**Fig 1. The times at which macular holes(MHs) were found or developed.**

developed during intraoperative subretinal t-PA injection in three eyes. MH developed post-operatively in one case. MH of this case was found during a second surgery for rebleeding from the RAM. In the initial surgery of the eye, t-PA was injected into the subretinal space from a small area of ILM peeling outside the fovea. The following day, rebleeding from the RAM occurred, and a preretinal hemorrhage was observed in the macula. The second surgery was performed seven days after the air injected into the vitreous in the initial surgery had dissipated, and the preretinal hemorrhage was removed, revealing the MH. Therefore, the exact date of MH occurrence is unknown, but is considered to be within seven days. No OCT data on MH exist because sub-ILM hemorrhage or vitreous hemorrhage prevented OCT examination on MH and because all MH were found or developed intraoperatively and closed postoperatively.

MH was found immediately after core vitrectomy in one case. MH was found during or immediately after ILM peeling in three eyes. MH developed during intraoperative subretinal t-PA injection in three eyes. MH developed postoperatively in one case.

The MH was closed by same surgical treatment that identified MH in all eight cases. Five eyes were treated by ILM peeling and gas tamponade, whereas two eyes were treated using an inverted ILM flap technique and gas tamponade. In the remaining eye, a small extrafoveal region of the ILM was removed, and the MH developed during subretinal injection of balanced salt solution. In this eye, gas tamponade was performed without increasing the extent of ILM peeling. The mean logMAR BCVA values were 0.98 (SD 0.51) in eyes with MHs and 1.11 (SD 0.47) in eyes without MHs; these did not significantly differ (p = 0.37).

None of the patients who received intravitreal t-PA injection preoperatively or intraoperatively developed MH.

In the RAM group, subretinal t-PA injection was performed in 12 eyes. In two (16.7%) of the 12 eyes, an MH developed before subretinal t-PA injection. In four (40%) of the remaining 10 eyes, an MH developed intraoperatively or postoperatively. In eyes without subretinal t-PA injection, no MHs developed after ILM peeling.

## Discussion

In this study, we compared 28 eyes with SMH due to RAM rupture and 19 eyes with SMH due to AMD. In this study, five MHs were found and three MHs developed intraoperatively in the RAM group. Eight of 28 eyes (28.6%) in the RAM group found or developed an MH, whereas no MHs occurred in the AMD group. There have been several reports of MHs in patients with SMH due to RAM rupture or SMH due to AMD [5–8,10,13–19], but no comparative studies have used the same methods at the same center. To our knowledge, this study is the first to directly compare the incidence of MH during or after vitrectomy between RAM and AMD groups at a single center.

Previous reports indicated that the incidences of MH associated with RAM rupture were 5-18% [18,20,21]. A previous meta-analysis revealed that the incidence of MH during PPV for RAM was 15% [22]. The reported incidences of MH in patients who underwent PPV for SMH due to RAM rupture were 9-50%. Although MH incidences vary among reports, they are expected to increase when study populations are limited to patients with SMH. In the present study, PPV was performed in patients with SMH due to RAM rupture; the total MH incidence was 8/28 (28.6%) and the MH incidence due to the RAM itself was 5/28 (17.9%), which are within the range of previous reports. In contrast, among previous reports concerning patients who underwent PPV for SMH due to AMD, MHs occurred in 1/36 (2.8%) [8] and 1/13 (7.7%) [17]. In the present study, there were no MHs among 19 patients. Published reports (including our study) indicate that the MH risk is much higher in eyes with SMH due to RAM rupture than in eyes with SMH due to AMD.

Several mechanisms have been proposed to explain the development of MHs in eyes with SMH. First, acute SMH might increase pressure in the subretinal space and lead to foveal dehiscence [19,20]. Second, tangential or anteroposterior vitreous traction into the fovea, due to VH-induced contraction of the posterior vitreous cortex, may contribute to MH development [5,7]. Third, persistent SMH causes retinal degeneration due to blood clot-derived fibrin, resulting in MH development [5,15,21]. Additionally, vitrectomy-induced mechanical damage to the macula can cause MH development [18,23]. If the timing of MH development was intraoperative or postoperative, the surgical procedures may have caused MH development. We speculate the main difference between RAM and AMD is the amount of subretinal pressure produced by the hemorrhage. The hemorrhage from RAM rupture originates from an artery and creates high subretinal pressure. The subretinal pressure generated by AMD-related hemorrhage is likely to be lower because this hemorrhage arises from choroidal neovascularization. Therefore, we suspect that subretinal pressure causes MHs more frequently in eyes with SMH due to RAM rupture than in eyes with SMH due to AMD.

Increased subretinal pressure caused by fluid injection may contribute to MH development. Our findings suggest that the incidence of MH is increased by subretinal t-PA injection for SMH due to RAM rupture. An MH developed during intraoperative subretinal t-PA injection in three eyes; an MH developed postoperatively in another eye that had undergone subretinal t-PA injection. Previous reports have described MH development in association with subretinal t-PA injection [16,18]. The previous findings are consistent with our results. MH has not occurred in patients with AMD who have undergone subretinal t-PA injection in this study. We consider that the risk of MH may be increased when subretinal t-PA infusion is performed in patients with RAM, because patients with SMH due to RAM have a weakened fovea, even if MH did not occur preoperatively. Although we speculate that subretinal fluid injection (including t-PA) increases the risk of MH development, it is difficult to draw a conclusion (based on this study alone) about whether subretinal t-PA injection should be used in surgical treatment of SMH due to RAM rupture; importantly, subretinal t-PA injection may improve SMH displacement and improve visual outcomes [24–26]. A larger prospective study is necessary to clearly establish the benefit of PPV with subretinal t-PA injection.

In this study, MHs were observed immediately after ILM peeling in three eyes. Even if an MH is not evident immediately after ILM peeling, the retina may become more fragile after ILM peeling, leading to increased incidence of MH. Kimura et al. reported a novel technique to prevent or treat MHs by creating a fissure near a sub-ILM hemorrhage outside of the central fovea and removing the sub-ILM hemorrhage from the fovea [27]. This technique may decrease the risk of MH development or may increase the likelihood of MH closure. Okanouchi et al. reported a technique for subretinal t-PA injection through a small area, in combination with ILM removal to decrease the injection pressure [28]. Although 12 psi was insufficient for subretinal t-PA injection into an area with intact ILM, subretinal injection through a small area was successfully performed using a pressure of 6 psi after ILM peeling. Subretinal injection with lower pressure might decrease the incidence of MH development, but this technique alone does not completely prevent MHs.

The present study is concerned with patients who have undergone PPV for SMH. To accurately assess the impact of PPV on MH, we decided to exclude patients who received other interventions before surgery; one patient in the RAM group and one patient in the AMD group received pneumatic treatment before surgery and were excluded from the study. The possibility cannot be excluded that this may have had an impact on the calculation of accurate MH incidence rates. This study may not accurately reflect the true incidence of the condition in all SMH patients, including those who were not scheduled for surgery.

In the present study, there was a relatively higher percentage of patients with VH in the AMD group (52.6%). Although pneumatic displacement with gas is often the preferred treatment for SMH due to AMD, PPV is required when VH is present. In our study, PPV was performed as initial treatment, so it is likely that a higher percentage of patients had VH.

There were several limitations in this study. First, it was a single-center, retrospective study with a small number of patients. Second, there were some differences in baseline characteristics (e.g., sex and age) between the RAM and AMD groups. Third, five surgeons participated in the study, and techniques such as ILM peeling or subretinal t-PA injection were performed at the surgeon's discretion. A prospective study with larger cases is necessary to clearly establish the relationship between surgical techniques and MH development. Fourth, many patients had VH or preretinal hemorrhage before surgery, and the SMH size and retinal thickness before surgery were not considered.

In conclusion, this study showed that MHs more frequently occurred during or after PPV in eyes with SMH caused by RAM rupture than in eyes with SMH caused by AMD. The incidence of MH may increase due to subretinal fluid injection during surgical treatment of SMH caused by RAM rupture. Surgeons performing vitrectomy for SMH due to RAM rupture should be aware of the potential for MH development.

## Supporting information

**S1 Dataset. All data included in the analysis.**
(XLSX)

## Acknowledgments

We thank Ryan Chastain-Gross, Ph.D., from Edanz (https://jp.edanz.com/ac) for editing a draft of this manuscript.

## Author contributions

**Conceptualization:** Riko Matsumoto, Masahito Ohji.

**Data curation:** Riko Matsumoto.

**Formal analysis:** Riko Matsumoto.

**Investigation:** Riko Matsumoto.

**Methodology:** Riko Matsumoto, Masashi Kakinoki, Osamu Sawada, Shumpei Obata, Yoshitsugu Saishin, Masahito Ohji.

**Project administration:** Riko Matsumoto.

**Supervision:** Masahito Ohji.

**Visualization:** Riko Matsumoto.

**Writing – original draft:** Riko Matsumoto.

**Writing – review & editing:** Masahito Ohji.

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
