## [Decision Letter · Decision Letter 0]

4 Oct 2024

PONE-D-24-29059Incidence of macular hole in patients undergoing pars plana vitrectomy for submacular hemorrhagePLOS ONE Dear Dr. Matsumoto,

Thank you for submitting your manuscript to PLOS ONE. After careful consideration, we feel that it has merit but does not fully meet PLOS ONE’s publication criteria as it currently stands. Therefore, we invite you to submit a revised version of the manuscript that addresses the points raised during the review process.

We look forward to receiving your revised manuscript.

Kind regards,

Jiro Kogo

Academic Editor

PLOS ONE

5. We note that there is identifying data in the Supporting Information file < PLOS ONE data.xlsx>. Due to the inclusion of these potentially identifying data, we have removed this file from your file inventory. Prior to sharing human research participant data, authors should consult with an ethics committee to ensure data are shared in accordance with participant consent and all applicable local laws.

-Location data

Reviewers' comments:

Reviewer's Responses to Questions

**Comments to the Author**

1. Is the manuscript technically sound, and do the data support the conclusions?

Reviewer #1: Partly

Reviewer #2: Yes

Reviewer #3: No

2. Has the statistical analysis been performed appropriately and rigorously? 

Reviewer #1: Yes

Reviewer #2: Yes

Reviewer #3: No

3. Have the authors made all data underlying the findings in their manuscript fully available?

Reviewer #1: Yes

Reviewer #2: Yes

Reviewer #3: Yes

4. Is the manuscript presented in an intelligible fashion and written in standard English?

Reviewer #1: No

Reviewer #2: Yes

Reviewer #3: Yes

5. Review Comments to the Author

Reviewer #1: This manuscript raises interesting concern about the development of MH in patients who underwent PPV for SMH. The authors described the differences in features between individuals with RAM and AMD who had been selected for the surgery. However, there are certain points should be clarified in order to maximize the manuscript's usefulness.

Methods:

Nowadays, treatment options for SMH vary, including intravitreal gas injections with and without TPA, and with and without anti-VEGF factor injections. It would be interesting to know about the indications and ocular features of the individuals that led the physician to recommend PPV for this group of patients, such as the size or duration of SMH (if available). Without this information, reporting on the incidence of MH-associated SMH among these operated patients may be biased.

Results:

1. Differences in pre and intra-operative interventions, such as timing of TPA injection (pre vs intra-operative injection), route of TPA injection (intravitreal vs subretinal injection), and intra-operative ILM peeling (done vs not done), are all potential risk factors for MH development. As a result, I recommend to add more details of these methods (either in text or in table form), as seen below.

• In patients who received a pre-operative intravitreal TPA injection, how many cases developed MH?

• In patients with ILM peeling and experienced MH, how many cases received subretinal gas/TPA injections?

• In patients that developed post-operative MH, how many days after surgery did it develop?

2. Regarding MH, it is interesting to know the difference in baseline features between RAM patients who acquired and did not develop MH. The information provided may be beneficial to the readers. I recommend that authors incorporate this information.

Discussion:

1. The authors should clarify that the incidence of MH in PPV patients may not accurately reflect the true incidence in all SMH patients which including those who did not schedule the operation.

2. The authors should add and emphasize that patients with pre-operative MH may not have the same mechanism as those who developed MH during or after vitrectomy, which could be due to disease pathology or surgical maneuvers.

3. 3. In the discussion of subretinal gas/TPA injection and MH, please further note that the volume and pressure might vary between patients due to retrospective studies. As a result, the relationship between subretinal TPA injection and MH should be investigated further in a prospective design.

Table 1:

Correct row for "preoperative VH"

Reviewer #2: I had the pleasure to review the manuscript by Matsumoto et al investigating the incidence of macular hole (MH) during pars plana vitrectomy for submacular hemorrhage due to either retinal arterial microaneurysm rupture or age-related macular degeneration. The manuscript is well formatted and fluent. The study design is clear and the introduction is well focused. The methods are widely explained and the results are clearly analyzed. The discussion is fluent , updated and solid.

Reviewer #3: Dear authors there appears to be a conflict trying to understand if the manuscript is focused on MH as a complication of SMH surgery or MH occuring after SMH in which case surgery (PPV) is not to be consider. You need to resolve this conflict or modify your title . Please se my comments below to provide some more information on your manuscript

Title : The title raises questions. First it gives the impression that the MH developed as a result of the surgery. But this is not the case as some eyes had MH even before the surgery, while other eyes developed the MH during or after the surgery. So this is a mixed group of MHs all related to the SMH and not necessarily to the surgery. The author needs to decide on if the surgery for SMH or the SMH is the focus of research.

Question: Were all the macular holes formed or noticed during vitreoretinal surgery? What of MHs that occurred post vitrectomy? How long after the primary surgery did the post operative MH occur? Were there extrafoveal MHs too?

The abstract’s purpose states that the MHs was during PPV for SMH. This does not appear to be correct and reflective of the work done as pre operative and post operative OCTs reflects timing of the MH as not only during surgery.

What does the author mean by” initial surgical treatment”?

What was the indication for the surgery and was present vitreous hemorrhage an indication for surgery in some cases? What technique was used for the surgery? Was it submacular surgery with subretinal tPA? Then a detailed description of the surgical procedure is required including the site of subretinal tPA injection.

Please include the mechanism of MH formation in the eyes with SMH?

Since this was a retrospective study, was there a standardized surgical procedure for all the cases.

Introduction section:

Again in line 47, “patients who underwent PPV” appears to mean that the focus is on the PPV for SMH.

Methods section:

Line 53: Why were patients who underwent pneumatic displacement before PPV excluded? This has a direct effect on the incidence. How many were excluded?

Line 63: Incidence of MH would not be one of the data extracted from the patients’ records. Rather MH and its dimensions on OCT would be a more likely data extracted.

Line 80: Which data was expressed as means and standard deviation?

Results section:

In table 1, the proportion of eyes with VH in the RAM group is missing.

I would have liked to see the OCT based size of the MHs including the minimum or basal diameter. I appreciate that this may not be available for all eyes since some eyes had a media opacity including VH.

It appears that subretinal t-PA carried a risk for the formation of intraoperative MH? Can the authors perform an analysis to examine the risk element comparing t-PA with the other grous.

Also, I would imagine that the height of the SMH would contribute to the mechanical stretching of the macular and cause considerable tangential and posterior anterior traction that could predispose to the development of the MH. Therefore can the author give an idea of the OCT based height of the SMH between the RAM and AMD eyes. More SMH and more height of elevated retina would be associated with more MH.

How many MHs occurred after vitrectomy? What was the time from vitrectomy to the occurrence of the MHs?

What was the rate of follow in the two groups?

A major drawback of this study is that multiple surgical techniques by multiple surgeons is likely to have been employed as this is a retrospective study. So it is truly difficult to compare and draw a reasonable conclusion since the surgical techniques are not controlled for. This general is a difficult study like any other surgical research in which the surgical techniques and steps are not controlled.

Discussion:

Because of the excluded eyes, the incidence is likely to be under estimated. Please provided the number of eyes excluded

6. PLOS authors have the option to publish the peer review history of their article (what does this mean? ). If published, this will include your full peer review and any attached files.

**Do you want your identity to be public for this peer review?** For information about this choice, including consent withdrawal, please see our Privacy Policy .

Reviewer #1: No

Reviewer #2: No

Reviewer #3: **Yes: ** OGUGUA NDUBUISI OKONKWO

---

## [Author Response · Author response to Decision Letter 1]

6 Nov 2024

Dear Dr. Kogo, and Reviewers,

Manuscript ID PONE-D-24-29059

Title: Incidence of macular hole in patients undergoing pars plana vitrectomy for submacular hemorrhage

Thank you very much for your careful review that contained important suggestions to improve our manuscript. We have studied the comments carefully and have made corrections, which we hope meet your approval.

Reviewer #1:

Methods: Nowadays, treatment options for SMH vary, including intravitreal gas injections with and without TPA, and with and without anti-VEGF factor injections. It would be interesting to know about the indications and ocular features of the individuals that led the physician to recommend PPV for this group of patients, such as the size or duration of SMH (if available). Without this information, reporting on the incidence of MH-associated SMH among these operated patients may be biased.

Our indications for surgery in SMH are cases complicated by VH, preretinal hemorrhage or sub-ILM hemorrhage and cases where the size of the SMH is approximately one disc diameter or larger. This statement has been added to Patients and Methods (page4, lines 59-61).

Results:

1. Differences in pre and intra-operative interventions, such as timing of TPA injection (pre vs intra-operative injection), route of TPA injection (intravitreal vs subretinal injection), and intra-operative ILM peeling (done vs not done), are all potential risk factors for MH development. As a result, I recommend to add more details of these methods (either in text or in table form), as seen below.

• In patients who received a pre-operative intravitreal TPA injection, how many cases developed MH?

The number of patients who received intravitreal t-PA injection was added to the results and table1. No cases developed MH (page 6, lines 105-107).

• In patients with ILM peeling and experienced MH, how many cases received subretinal gas/TPA injections?

ILM peeling involving the fovea was performed in six cases of the eight cases that developed MH. Subretinal t-PA injection was performed in four cases of these six cases. (page7, lines123-125).

• In patients that developed post-operative MH, how many days after surgery did it develop?

MH developed postoperatively in one case. MH of this case was found during a second surgery for rebleeding from the RAM. In the initial surgery of the eye, t-PA was injected into the subretinal space from a small area of ILM peeling outside the fovea. The following day, rebleeding from the RAM occurred, and a preretinal hemorrhage was observed in the macula. The second surgery was performed seven days after the air injected into the vitreous in the initial surgery had dissipated, and the preretinal hemorrhage was removed, revealing the MH. Therefore, the exact date of MH occurrence is unknown, but is considered to be within seven days. (page 7, line 129-136).

2. Regarding MH, it is interesting to know the difference in baseline features between RAM patients who acquired and did not develop MH. The information provided may be beneficial to the readers. I recommend that authors incorporate this information.

In accordance with the reviewer's comment, we have added Table2 (page8).

Discussion:

1. The authors should clarify that the incidence of MH in PPV patients may not accurately reflect the true incidence in all SMH patients which including those who did not schedule the operation.

In accordance with the reviewer's comment, we have added the following sentence into the Discussion. “This study may not accurately reflect the true incidence of the condition in all SMH patients, including those who were not scheduled for surgery.” (page11, line 221-222)

2. The authors should add and emphasize that patients with pre-operative MH may not have the same mechanism as those who developed MH during or after vitrectomy, which could be due to disease pathology or surgical maneuvers.

In accordance with the reviewer's comment, we have added the following sentence into the Discussion. “If the timing of MH development was intraoperative or postoperative, the surgical procedures may have caused MH development.” (page 10, line 185-186)

3. In the discussion of subretinal gas/TPA injection and MH, please further note that the volume and pressure might vary between patients due to retrospective studies. As a result, the relationship between subretinal TPA injection and MH should be investigated further in a prospective design.

In accordance with the reviewer's comment, we have added the following sentence into the Discussion. “A prospective study with larger cases is necessary to clearly establish the relationship between surgical techniques and MH development.” (page 12, line 231-232)

Table 1: Correct row for "preoperative VH"

We corrected this.

Reviewer #2:

I had the pleasure to review the manuscript by Matsumoto et al investigating the incidence of macular hole (MH) during pars plana vitrectomy for submacular hemorrhage due to either retinal arterial microaneurysm rupture or age-related macular degeneration. The manuscript is well formatted and fluent. The study design is clear and the introduction is well focused. The methods are widely explained and the results are clearly analyzed. The discussion is fluent, updated and solid.

We greatly appreciate your comment.

Reviewer #3

Dear authors there appears to be a conflict trying to understand if the manuscript is focused on MH as a complication of SMH surgery or MH occuring after SMH in which case surgery (PPV) is not to be consider. You need to resolve this conflict or modify your title. Please my comments below to provide some more information on your manuscript

Title : The title raises questions. First it gives the impression that the MH developed as a result of the surgery. But this is not the case as some eyes had MH even before the surgery, while other eyes developed the MH during or after the surgery. So this is a mixed group of MHs all related to the SMH and not necessarily to the surgery. The author needs to decide on if the surgery for SMH or the SMH is the focus of research.

All MHs that occurred in this study were discovered intraoperatively; we emphasize that all MHs were discovered intraoperatively to make it clear that the purpose of this study was to examine the impact of surgery on SMH. (page 7, line 121)

Question: Were all the macular holes formed or noticed during vitreoretinal surgery? What of MHs that occurred post vitrectomy? How long after the primary surgery did the post operative MH occur? Were there extrafoveal MHs too?

All MHs were found or developed during surgery. MH developed postoperatively in one case. MH of this case was found during a second surgery for rebleeding from the RAM. In the initial surgery of the eye, t-PA was injected into the subretinal space from a small area of ILM peeling outside the fovea. The following day, rebleeding from the RAM occurred, and a preretinal hemorrhage was observed in the macula. The second surgery was performed seven days after the air injected into the vitreous in the initial surgery had dissipated, and the preretinal hemorrhage was removed, revealing the MH. Therefore, the exact date of MH occurrence is unknown, but is considered to be within seven days. (page 7, line 129-136). There was no extrafoveal MHs.

The abstract’s purpose states that the MHs was during PPV for SMH. This does not appear to be correct and reflective of the work done as pre operative and post operative OCTs reflects timing of the MH as not only during surgery.

In this study, all MHs were found or developed during surgery. The phrase “by preoperative or postoperative optical coherence tomography” was removed from the abstract.

What does the author mean by” initial surgical treatment”?

“Initial surgical treatment” refers to the first surgery if the patient has had multiple surgeries.

What was the indication for the surgery and was present vitreous hemorrhage an indication for surgery in some cases?

This includes cases in which SMH was found during surgery because fundus examination could not be performed before surgery due to vitreous hemorrhage.

What technique was used for the surgery? Was it submacular surgery with subretinal tPA? Then a detailed description of the surgical procedure is required including the site of subretinal tPA injection.

We have added a description of the surgical technique to the METHODS section. The surgical technique for subretinal injection of t-PA is as follows: the ILM at the planned t-PA injection site was peeled, and BSS was injected into the subretinal space using a micro cannula, after which 0.1 ml (25μg) of t-PA was injected. BSS was injected before tPA injection to inject the exact amount of tPA. The injection pressure was controlled using a viscous fluid control system (Alcon Laboratories, FortWorth, TX). (page 4, line 70-74)

Please include the mechanism of MH formation in the eyes with SMH?

The mechanism of MH formation in SMH is described in Discussion. (page 10, line179-186)

Since this was a retrospective study, was there a standardized surgical procedure for all the cases.

Since this is a retrospective study and the duration of the study is as long as 13 years, there is no standardized surgical procedure for all cases. The surgical procedure was determined at the discretion of the surgeons, and this is noted in the limitations. (page 12 line 229-23)

Introduction section:

Again in line 47, “patients who underwent PPV” appears to mean that the focus is on the PPV for SMH.

This study focuses on patients who received PPV for SMH.

Line 53: Why were patients who underwent pneumatic displacement before PPV excluded? This has a direct effect on the incidence. How many were excluded?

To accurately assess the impact of PPV on MH, we decided to exclude patients who received other interventions before surgery; one patient in the RAM group and one patient in the AMD group received pneumatic treatment before surgery and were excluded from the study. (page 11, line 216-222)

Line 63: Incidence of MH would not be one of the data extracted from the patients’ records. Rather MH and its dimensions on OCT would be a more likely data extracted.

The phrase “incidence of MH” was changed to “Mention of MH in the surgical records” in response to the reviewer's comment (page 4, line 65). No OCT data on MH exist because sub-ILM hemorrhage or vitreous hemorrhage prevented OCT examination on MH and because all MH were found or developed intraoperatively and closed postoperatively.

Line 80: Which data was expressed as means and standard deviation?

In accordance with the reviewer's comment, we have added to the Methods which data it is.

Results section:

In table 1, the proportion of eyes with VH in the RAM group is missing.

We corrected this.

I would have liked to see the OCT based size of the MHs including the minimum or basal diameter. I appreciate that this may not be available for all eyes since some eyes had a media opacity including VH.

No OCT data on MH exist because all MH were found or developed intraoperatively and closed postoperatively.

It appears that subretinal t-PA carried a risk for the formation of intraoperative MH? Can the authors perform an analysis to examine the risk element comparing t-PA with the other grous.

As the reviewer has noted, there is a potential risk of intraoperative MH formation associated with subretinal injection of t-PA. However, it is regrettable that further analysis cannot be conducted in the present study.

Also, I would imagine that the height of the SMH would contribute to the mechanical stretching of the macular and cause considerable tangential and posterior anterior traction that could predispose to the development of the MH. Therefore can the author give an idea of the OCT based height of the SMH between the RAM and AMD eyes. More SMH and more height of elevated retina would be associated with more MH.

As the reviewer asserted, we concur that the height of SMH based on OCT is a valuable indicator for investigating the predisposition for the development of MH. However, in the present study, the preoperative SMH height could be assessed in only a few cases due to the presence of sub-ILM hemorrhage or VH. Only 11 of the 28 cases in the RAM group and 3 of the 19 cases in the AMD group had SMH height that was measurable by OCT. Consequently, the paper did not include information on SMH height by OCT.

How many MHs occurred after vitrectomy? What was the time from vitrectomy to the occurrence of the MHs?

One MH occurred after vitrectomy. MH of this case was found during a second surgery for rebleeding from the RAM. In the initial surgery of the eye, t-PA was injected into the subretinal space from a small area of ILM peeling outside the fovea. The following day, rebleeding from the RAM occurred, and a preretinal hemorrhage was observed in the macula. The second surgery was performed seven days after the air injected into the vitreous in the initial surgery had dissipated, and the preretinal hemorrhage was removed, revealing the MH. (page 7, line 129-136)

What was the rate of follow in the two groups?

The mean and median postoperative observation periods for the RAM group were found to be 13.3 and 12 months, respectively. The mean and median postoperative observation periods for the AMD group were found to be 40.9 and 37 months, respectively. (page 6, line 110-113)

A major drawback of this study is that multiple surgical techniques by multiple surgeons is likely to have been employed as this is a retrospective study. So it is truly difficult to compare and draw a reasonable conclusion since the surgical techniques are not controlled for. This general is a difficult study like any other surgical research in which the surgical techniques and steps are not controlled.

As the reviewer pointed out, one of the shortcomings of this study is the lack of control over surgical techniques and procedures. We have added the following language to our discussion:” A prospective study with larger number of cases is necessary to clearly establish the relationship between surgical techniques and MH.”

Discussion:

Because of the excluded eyes, the incidence is likely to be under estimated. Please provided the number of eyes excluded

One patient in the RAM group and one patient in the AMD group received pneumatic treatment before surgery and were excluded from the study. The possibility cannot be excluded that this may have had an impact on the calculation of accurate MH incidence rates. We have added this to our results and discussion. (page 5 line 93-95, page 11 line 216-222)

We greatly appreciate both your help and that of the referees concerning improvements to this paper. I hope that the revised manuscript is now suitable for publication.

Sincerely yours,

Riko Matsumoto, MD, corresponding author

Department of Ophthalmology, Shiga University of Medical Science

Seta-Tsukinowacho, Otsu, Shiga 520-2192, JAPAN

Tel: +81-77-548-2276

Fax: +81-77-548-2279

e-mail: rokamoto@belle.shiga-med.ac.jp

---

## [Decision Letter · Decision Letter 1]

29 Nov 2024

PONE-D-24-29059R1Incidence of macular hole in patients undergoing pars plana vitrectomy for submacular hemorrhagePLOS ONE

Dear Dr. Matsumoto,

Thank you for submitting your manuscript to PLOS ONE. After careful consideration, we feel that it has merit but does not fully meet PLOS ONE’s publication criteria as it currently stands. Therefore, we invite you to submit a revised version of the manuscript that addresses the points raised during the review process.

We look forward to receiving your revised manuscript.

Kind regards,

Jiro Kogo

Academic Editor

PLOS ONE

Reviewers' comments:

Reviewer's Responses to Questions

**Comments to the Author**

1. If the authors have adequately addressed your comments raised in a previous round of review and you feel that this manuscript is now acceptable for publication, you may indicate that here to bypass the “Comments to the Author” section, enter your conflict of interest statement in the “Confidential to Editor” section, and submit your "Accept" recommendation.

Reviewer #1: All comments have been addressed

Reviewer #3: (No Response)

2. Is the manuscript technically sound, and do the data support the conclusions?

Reviewer #1: Yes

Reviewer #3: Partly

3. Has the statistical analysis been performed appropriately and rigorously? 

Reviewer #1: Yes

Reviewer #3: Yes

4. Have the authors made all data underlying the findings in their manuscript fully available?

Reviewer #1: Yes

Reviewer #3: Yes

5. Is the manuscript presented in an intelligible fashion and written in standard English?

Reviewer #1: Yes

Reviewer #3: Yes

6. Review Comments to the Author

Reviewer #1: I would like to express my gratitude to the authors for their substantial contribution to the scientific knowledge conveyed in this study. Additionally, the limitations related to the retrospective data analysis have been emphasized more clearly. I have a few minor recommendations for the authors.

Introduction and abstract and Table:

- It is advisable to use the genera term "rupture arterial macroaneurysm" in place of " rupture arterial microaneurysm."

Results:

Line: 105-107

“Intravitreal t-PA injection was performed in three eyes (10.7%, day before surgery: 2, in surgery: 1) in the RAM group and two eyes (10.5%, 107 before surgery: 2) in the AMD group (p=1).”

This statement presents challenges in interpretation. It is advisable to eliminate either surgery 1 or surgery 2 from this statement and describe using an alternative sentence.

Line 155:

“None of the patients who received intravitreal t-PA injection developed MH.”

It would be more accurate to state the time point of injection, (preoperative, correct?).

Table 1 and Table 2:

Considering use:

“Preoperative intravitreal t-PA” instead of “intravitreal t-PA”.

“Intraoperative subretinal t-PA injection” instead of “subretinal t-PA injection”

It is advisable to include the Mean ± SD within parentheses for continuous variables.

Reviewer #3: Dear Authors, it is my impression that you have put in a lot of work to revise this manuscript. However, I still have difficulties getting a clear understanding of the reporting of your research.

For example, you have stated that all the macular holes were discovered intraoperatively, this is not clearly reflected in the abstract.

Please state clearly that in all cases the MH was “discovered” during PPV after removal of the vitreous hemorrhage. This helps explain why the MH was identified only during PPV.

Secondly, following your explanation in the revised letter, the incidence can not include those cases in which MH was induced by the surgical procedure or subretinal TPA injection. To give the true incidence, the sample MUST be representative of only those cases that was as a result of RAP or nAMD only and not iatrogenic. Therefore those MHs that developed following surgery have to be excluded all through the manuscript.

“Purpose: To investigate the incidence of macular hole (MH) during pars plana vitrectomy (PPV) for submacular hemorrhage (SMH) due to either retinal arterial microaneurysm (RAM) rupture or age-related macular degeneration (AMD). Methods: We retrospectively evaluated 47 eyes of 47 patients with SMH due to RAM rupture or AMD who underwent PPV. The presence or development of MHs was confirmed intraoperatively by the surgeon. We compared the incidence of MH between the RAM and AMD groups

This understanding is also lacking in the later segment below

“The MH was closed by initial surgical treatment in all eight cases. In the RAM group, subretinal tissue plasminogen activator (t-PA) injection was performed in 12 eyes. Of the 12 eyes, two developed an MH before t-PA subretinal injection. Of the remaining 10 eyes, four (40%) developed an MH intraoperatively or postoperatively”

First, please make it clear that the MH was closed during the same procedure to remove the vitreous hemorrhage. Initial surgical treatment gives the impression that a planned MH surgery was done.

Secondly, those eyes that developed MH after t-PA injection should be excluded for the simple reason that the intervention could be the reason for MH and not entirely the disease. So, if the injection was not done, the MH may not have occurred. This point has to be reflected in the manuscript.

7. PLOS authors have the option to publish the peer review history of their article (what does this mean? ). If published, this will include your full peer review and any attached files.

**Do you want your identity to be public for this peer review?** For information about this choice, including consent withdrawal, please see our Privacy Policy .

Reviewer #1: No

Reviewer #3: **Yes: ** OGUGUA NDUBUISI OKONKWO

---

## [Author Response · Author response to Decision Letter 2]

8 Jan 2025

Response to Reviewers letter

Dr. Jiro Kogo

Dear Dr. Kogo, and Reviewers,

Manuscript ID PONE-D-24-29059

Title: Incidence of macular hole in patients undergoing pars plana vitrectomy for submacular hemorrhage

Thank you very much for your careful review that contained important suggestions to improve our manuscript. We have studied the comments carefully and have made corrections, which we hope meet your approval.

Reviewers’ comments are reproduced and are followed by our responses.

Reviewer #1

I would like to express my gratitude to the authors for their substantial contribution to the scientific knowledge conveyed in this study. Additionally, the limitations related to the retrospective data analysis have been emphasized more clearly. I have a few minor recommendations for the authors.

Introduction and abstract and Table:

- It is advisable to use the genera term "rupture arterial macroaneurysm" in place of " rupture arterial microaneurysm."

We corrected this term.

Results:

Line: 105-107

“Intravitreal t-PA injection was performed in three eyes (10.7%, day before surgery: 2, in surgery: 1) in the RAM group and two eyes (10.5%, 107 before surgery: 2) in the AMD group (p=1).”

This statement presents challenges in interpretation. It is advisable to eliminate either surgery 1 or surgery 2 from this statement and describe using an alternative sentence.

The sentence pointed out (Line: 105-107) was eliminated, and the following sentences were added. “In the RAM group, intravitreal t-PA injection was performed in three eyes (10.7%); the day before surgery in two eyes, and during surgery in the other eye. In the AMD group, intravitreal t-PA injection was performed in two eyes (10.5%) the day before surgery.”

Line 155:

“None of the patients who received intravitreal t-PA injection developed MH.”

It would be more accurate to state the time point of injection, (preoperative, correct?).

We have added the phrase “preoperatively or intraoperatively” to the sentence. (page9, line159)

Table 1 and Table 2:

Considering use:

“Preoperative intravitreal t-PA” instead of “intravitreal t-PA”.

“Intraoperative subretinal t-PA injection” instead of “subretinal t-PA injection”

We could not make this change because “intravitreal t-PA injection” is included both intraoperatively and preoperatively.

It is advisable to include the Mean ± SD within parentheses for continuous variables.

In accordance with the reviewer's comment, the notation of the standard deviation has been changed.

Reviewer #3

Dear Authors, it is my impression that you have put in a lot of work to revise this manuscript. However, I still have difficulties getting a clear understanding of the reporting of your research.

For example, you have stated that all the macular holes were discovered intraoperatively, this is not clearly reflected in the abstract.

Please state clearly that in all cases the MH was “discovered” during PPV after removal of the vitreous hemorrhage. This helps explain why the MH was identified only during PPV.

We added in the abstract that all MH were found intraoperatively.

Secondly, following your explanation in the revised letter, the incidence can not include those cases in which MH was induced by the surgical procedure or subretinal TPA injection. To give the true incidence, the sample MUST be representative of only those cases that was as a result of RAP or nAMD only and not iatrogenic. Therefore those MHs that developed following surgery have to be excluded all through the manuscript.

“Purpose: To investigate the incidence of macular hole (MH) during pars plana vitrectomy (PPV) for submacular hemorrhage (SMH) due to either retinal arterial microaneurysm (RAM) rupture or age-related macular degeneration (AMD). Methods: We retrospectively evaluated 47 eyes of 47 patients with SMH due to RAM rupture or AMD who underwent PPV. The presence or development of MHs was confirmed intraoperatively by the surgeon. We compared the incidence of MH between the RAM and AMD groups

This understanding is also lacking in the later segment below

“The MH was closed by initial surgical treatment in all eight cases. In the RAM group, subretinal tissue plasminogen activator (t-PA) injection was performed in 12 eyes. Of the 12 eyes, two developed an MH before t-PA subretinal injection. Of the remaining 10 eyes, four (40%) developed an MH intraoperatively or postoperatively”

First, please make it clear that the MH was closed during the same procedure to remove the vitreous hemorrhage. Initial surgical treatment gives the impression that a planned MH surgery was done.

The sentence “The MH was closed by initial surgical treatment in all eight cases.” was changed to “The MH was closed by same surgical treatment that identified MH in all eight cases.” in response to the reviewer's comment (page8, line152). Abstract was also changed.

Secondly, those eyes that developed MH after t-PA injection should be excluded for the simple reason that the intervention could be the reason for MH and not entirely the disease. So, if the injection was not done, the MH may not have occurred. This point has to be reflected in the manuscript.

Thank you for your suggestion. You have raised an important point; however, we believe that while the incidence of MH due to the disease itself is important, another important message is that subretinal t-PA injection causes MH in RAM, but not in AMD, and therefore requires attention during surgery. We have divided the incidence of MH in the results into those due to the disease itself and those related to the surgical technique in the result (page7 , line121-124 ).We also have added the following text in the discussion to indicate the need for caution with subretinal t-PA injection, particularly in RAM rather than AMD. ” MH has not occurred in patients with AMD who have undergone subretinal t-PA injection in this study. We consider that the risk of MH may be increased when subretinal t-PA infusion is performed in patients with RAM, because patients with SMH due to RAM have a weakened fovea, even if MH did not occur preoperatively. “(page11 , line206-210)

We greatly appreciate both your help and that of the referees concerning improvements to this paper. I hope that the revised manuscript is now suitable for publication.

Sincerely yours,

Riko Matsumoto, MD, corresponding author

Department of Ophthalmology, Shiga University of Medical Science

Seta-Tsukinowacho, Otsu, Shiga 520-2192, JAPAN

Tel: +81-77-548-2276

Fax: +81-77-548-2279

e-mail: rokamoto@belle.shiga-med.ac.jp

---

## [Decision Letter · Decision Letter 2]

30 Jan 2025

Incidence of macular hole in patients undergoing pars plana vitrectomy for submacular hemorrhage

PONE-D-24-29059R2

Dear Dr. Matsumoto

We’re pleased to inform you that your manuscript has been judged scientifically suitable for publication and will be formally accepted for publication once it meets all outstanding technical requirements.

Kind regards,

Jiro Kogo

Academic Editor

PLOS ONE

Additional Editor Comments (optional):

Reviewers' comments:

Reviewer's Responses to Questions

**Comments to the Author**

1. If the authors have adequately addressed your comments raised in a previous round of review and you feel that this manuscript is now acceptable for publication, you may indicate that here to bypass the “Comments to the Author” section, enter your conflict of interest statement in the “Confidential to Editor” section, and submit your "Accept" recommendation.

Reviewer #1: All comments have been addressed

Reviewer #3: All comments have been addressed

2. Is the manuscript technically sound, and do the data support the conclusions?

Reviewer #1: Yes

Reviewer #3: Yes

3. Has the statistical analysis been performed appropriately and rigorously? 

Reviewer #1: Yes

Reviewer #3: Yes

4. Have the authors made all data underlying the findings in their manuscript fully available?

Reviewer #1: Yes

Reviewer #3: Yes

5. Is the manuscript presented in an intelligible fashion and written in standard English?

Reviewer #1: Yes

Reviewer #3: Yes

6. Review Comments to the Author

Reviewer #1: This manuscript explores the incidence of MH found during the operation for SMH which is one point that should be considered. Several points have been responded and the revised manuscript has been improved.

Reviewer #3: The authors have addressed the issues raised in previous rounds of review satisfactorily.

This paper can now be accepted for publication.

7. PLOS authors have the option to publish the peer review history of their article (what does this mean? ). If published, this will include your full peer review and any attached files.

**Do you want your identity to be public for this peer review?** For information about this choice, including consent withdrawal, please see our Privacy Policy .

Reviewer #1: No

Reviewer #3: **Yes: ** OGUGUA NDUBUISI OKONKWO

---

## [Editor Report · Acceptance letter]

PONE-D-24-29059R2

PLOS ONE

Dear Dr. Matsumoto,

I'm pleased to inform you that your manuscript has been deemed suitable for publication in PLOS ONE. Congratulations! Your manuscript is now being handed over to our production team.

Kind regards,

on behalf of

Prof. Jiro Kogo

Academic Editor

PLOS ONE